# Taste shaped the use of botanical drugs

Marco Leonti[1]*, Joanna Baker[2]*, Peter Staub[1], Laura Casu[3], Julie Hawkins[2]

[1]Department of Biomedical Sciences, University of Cagliari, Cittadella Universitaria, Monserrato, Italy; [2]School of Biological Sciences, University of Reading, Reading, United Kingdom; [3]Department of Life and Environmental Sciences, University of Cagliari, Cittadella Universitaria, Monserrato, Italy

**Abstract** The perception of taste and flavour (a combination of taste, smell, and chemesthesis), here also referred to as chemosensation, enables animals to find high-value foods and avoid toxins. Humans have learned to use unpalatable and toxic substances as medicines, yet the importance of chemosensation in this process is poorly understood. Here, we generate tasting-panel data for botanical drugs and apply phylogenetic generalised linear mixed models to test whether intensity and complexity of chemosensory qualities as well as particular tastes and flavours can predict ancient Graeco-Roman drug use. We found chemosensation to be strongly predictive of therapeutic use: botanical drugs with high therapeutic versatility have simple yet intense tastes and flavours, and 21 of 22 chemosensory qualities predicted at least one therapeutic use. In addition to the common notion of bitter tasting medicines, we also found starchy, musky, sweet, and soapy drugs associated with versatility. In ancient Greece and Rome, illness was thought to arise from imbalance in bodily fluids or humours, yet our study suggests that uses of drugs were based on observed physiological effects that are often consistent with modern understanding of chemesthesis and taste receptor pharmacology.

*For correspondence:
mleonti@unica.it (ML);
j.l.a.baker@reading.ac.uk (JB)

Competing interest: The authors declare that no competing interests exist.

## eLife assessment

This **valuable** study links the "taste" of botanicals to their application as medicines used by the ancient Greco-Roman society. The authors used phylogenetic linear mixed models in a Bayesian framework to test the relationships between taste qualities, intensities, complexities, and therapeutic use. The evidence supporting the conclusions is **solid**, although there is a minor weakness concerning the somewhat inconsistent method of botanical preparation and presentation to the taster panelists; subjective bias and robustness of the participants' responses might have been overlooked. The study may be of broad interest to pharmacologists and scientists working on drug discovery, particularly those interested in natural products.

## Introduction

'*A bitter pill to swallow*' is just one of many common idioms referring to the taste of medicine, which from ancient to early modern times has relied mostly on botanical drugs (**Touwaide, 2022**). In ancient Greece and Rome, the expected effects of botanical drugs were explained by their taste (**Jones, 1959**; **Einarson and Link, 1990**; **Jouanna, 2012**) and according to the prevailing medical theory, disease resulted from a disequilibrium between bodily fluids or humours including phlegm, blood, yellow bile, and black bile; drugs affected the flow and balance of these humours (**Jones, 1959**; **King, 2013**). Theophrastus (300 BCE) attributed sweet the capacity to smoothen, astringent with the power to desiccate and solidify, pungent with the capacity to cut or to separate out the heat, salty with desiccating and irritating properties, and bitter with the capacity to melt and irritate (**Einarson and Link, 1990**). Other systems of medicine, such as Ayurveda and Traditional Chinese Medicine, continue

**eLife digest** In ancient times people used trial and error to identify medicinal plants as being effective. Later, diseases were believed to arise from imbalances in body fluids (or 'humours'), and botanical drugs were thought to restore this balance through the power of their taste. Modern science rejects this theory but does recognise the importance of chemosensation – our sensitivity to chemicals through taste and smell. These senses evolved in humans to help us seek out nutrients and avoid toxins and may also have guided the ancient uses of botanical drugs.

There are many records of historical medicinal plant use and ailments, which makes it possible to explore possible relationships between therapeutic uses of botanical drugs and their chemosensory qualities. To investigate if therapeutic uses of botanical drugs could indeed be predicted by taste and flavour, Leonti, Baker et al. collected 700 botanical drugs identified in an ancient text, named *De Materia Medica*, which dates back to the 1st century CE.

The researchers asked volunteer tasters to classify the botanical drugs using 22 taste descriptions, such as bitter, aromatic, burning/hot, and fresh/cooling. The volunteers were also asked to score the strength of these tastes. Leonti, Baker et al. then used statistical modelling to see if the participant's taste descriptions could be used to predict the therapeutic uses of the drugs identified in the ancient text.

This revealed that of the 46 therapeutic indications described in the text, 45 showed significant associations with at least one taste quality. Botanical drugs with stronger and simpler tastes tended to be used for a wider range of therapeutic indications. This suggests that chemosensation influenced therapeutic expectations in ancient, prescientific medicine.

The study of Leonti, Baker et al. brings ancient medicine to life, offering valuable insights into the chemosensory aspects of medicinal plants and their potential applications in modern medicine. A next step would be to explore whether these insights could have relevance to modern science.

to classify *materia medica* according to tastes and flavours (*Dragos and Gilca, 2018*; *Shou-zhong, 1998*), and indigenous societies use chemosensory qualities to select botanical drugs (*Casagrande, 2000*; *Leonti et al., 2002*; *Shepard, 2004*).

The perception of taste and flavour (a combination of taste, smell, and chemesthesis), here also referred to as chemosensation, has evolved to meet nutritional requirements and is particularly important in omnivores for seeking out nutrients and avoiding toxins (*Rozin and Todd, 2016*; *Breslin, 2013*; *Glendinning, 2022*). The rejection of bitter stimuli has generally been associated with the avoidance of toxins (*Glendinning, 1994*; *Lindemann, 2001*; *Breslin, 2013*) but to date no clear relationship between bitter compounds and toxicity at a nutritionally relevant dose could be established (*Glendinning, 1994*; *Nissim et al., 2017*). While bitter tasting metabolites occurring in fruits and vegetables have been linked with a lower risk for contracting cancer and cardiovascular diseases (*Drewnowski and Gomez-Carneros, 2000*), the avoidance of pharmacologically active compounds is probably the reason why many medicines, including botanical drugs, taste bitter (*Johns, 1990*; *Mennella et al., 2013*).

Botanical drugs are chemically complex and often used for a range of different health problems (*Wagner et al., 2007*). Chemical complexity determines pharmacological complexity, since different chemicals interact with different targets (*Gertsch, 2011*), and influence taste complexity and intensity (*Spence and Wang, 2018*; *Breslin and Beauchamp, 1997*; *Green et al., 2010*). Today, some associations between chemosensory qualities and therapeutic uses, such as bitter-tasting botanical drugs for stomach problems or astringent ones for diarrhoea, are still recognised in Western herbal medicine (*Wagner et al., 2007*). These observations establish a role for chemosensory qualities in medicine. Yet, taste and flavour are largely irrelevant to modern medicine, aside from the pharmaceutical industry looking for ways to mask bad tastes (*Mennella et al., 2013*).

Whether ancient systems of traditional medicine could be relevant to modern science is questionable. The theory of humours was the principal underlying the concept of Western medicine from ancient Graeco-Roman times until the Industrial Age, but as an entirely discredited theory it might be expected that associated practices also lack scientific basis.

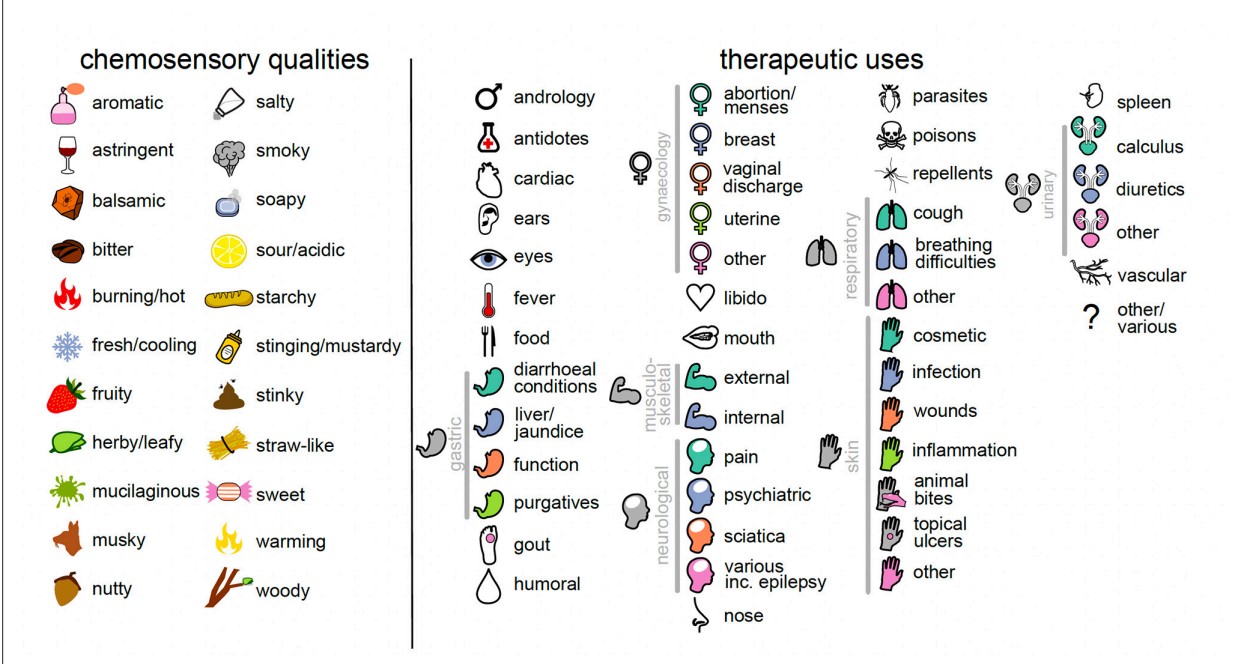

**Figure 1.** The 22 chemosensory qualities and 46 therapeutic uses studied here. Each chemosensory quality and use is represented by an icon that is used throughout the manuscript. Therapeutic uses that share an icon are considered to represent the same category of use (25 in total); these are linked by grey bars.

However, we propose that taste and flavour perception of botanical drugs links humoral theory to modern medicine. We hypothesise that chemosensation and underlying physiological effects mediated by chemesthesis and taste receptor pharmacology co-explain the diversity of therapeutic uses of botanical drugs in ancient times. We further hypothesise that associations between perceived complexity and intensity of chemosensory qualities of botanical drugs and their versatility (number of categories of therapeutic use a botanical drug is recommended for) provide insights into the development of empirical pharmacological knowledge and therapeutic behaviour. If tastes and flavours, as determined by a modern-day tasting panel, do predict uses as recorded in ancient texts, this would be an unprecedented insight into how chemosensation in combination with physiological effects guided human behaviour in pre-historic times and conditioned therapeutic expectations and humoral theory in historic times. To test these hypotheses, we assessed the relationships between chemosensory qualities, their intensity, and complexity with therapeutic uses of botanical drugs described in Pedanios Dioscorides' pharmacognostic compendium, *De Materia Medica* (*DMM*; *Matthioli, 1970*; *Staub et al., 2016*), the foremost pharmaceutical source of antiquity. *DMM* compiles knowledge on the sourcing, preparation, and therapeutic consensus of botanical drugs traded and used in the Eastern Mediterranean region of the Roman Empire during the first century CE (*Riddle, 1985*). We collected 700 botanical drugs described in *DMM* (*Supplementary file 1*) and used quality descriptors to represent chemosensation by a trained tasting panel scoring each botanical drug for presence and intensity of each quality (*Figure 1*; *Supplementary files 2 and 3*). We arranged the botanical drugs into therapeutic uses according to affected organs, therapeutic functions, diseases, and symptoms as indicated in *DMM*, and into broader and more inclusive categories of therapeutic use (*Figure 1*; *Supplementary files 1 and 4*). By applying phylogenetic generalised linear mixed models (PGLMMs) to our data and a plant phylogeny (*Zanne et al., 2014*; see Methods), we tested whether perceived taste and flavour qualities, as well as their overall intensity and complexity predict recorded therapeutic uses whilst simultaneously accounting for the confounding effects of the shared uses and chemosensory stimuli by drugs from closely related species.

## Results

### Diversity of chemosensory qualities and therapeutic uses

We used 22 descriptors to represent perceived chemosensory qualities of 700 botanical drugs and identified 46 therapeutic uses across 25 categories as described in DMM (*Figure 1*; *Supplementary files 1 and 4*). A group of 11 panellists conducted 3973 sensory trials (see Methods) and reported 10,463 individual perceptions of qualities. The most frequently reported qualities were bitter (1556 reports; 39% of all assessed samples), herby/leafy (1146 reports; 29%), aromatic (795 reports; 20%), and stinging (730 reports; 18%), while the least was mucilaginous (128 reports; 3.2%). Weak qualities were most frequently reported (5244 reports; 50%); 3899 (37%) of reports were of medium strength and 1320 (13%) were perceived as strong. Balsamic (69 reports; 24%), burning/hot (41 reports; 22%), and fresh/cooling (49 reports; 21%) were the qualities most often perceived as strong, while smoky (87 reports; 61%), sour (224 reports; 60%), and starchy (190 reports; 59%) the qualities most

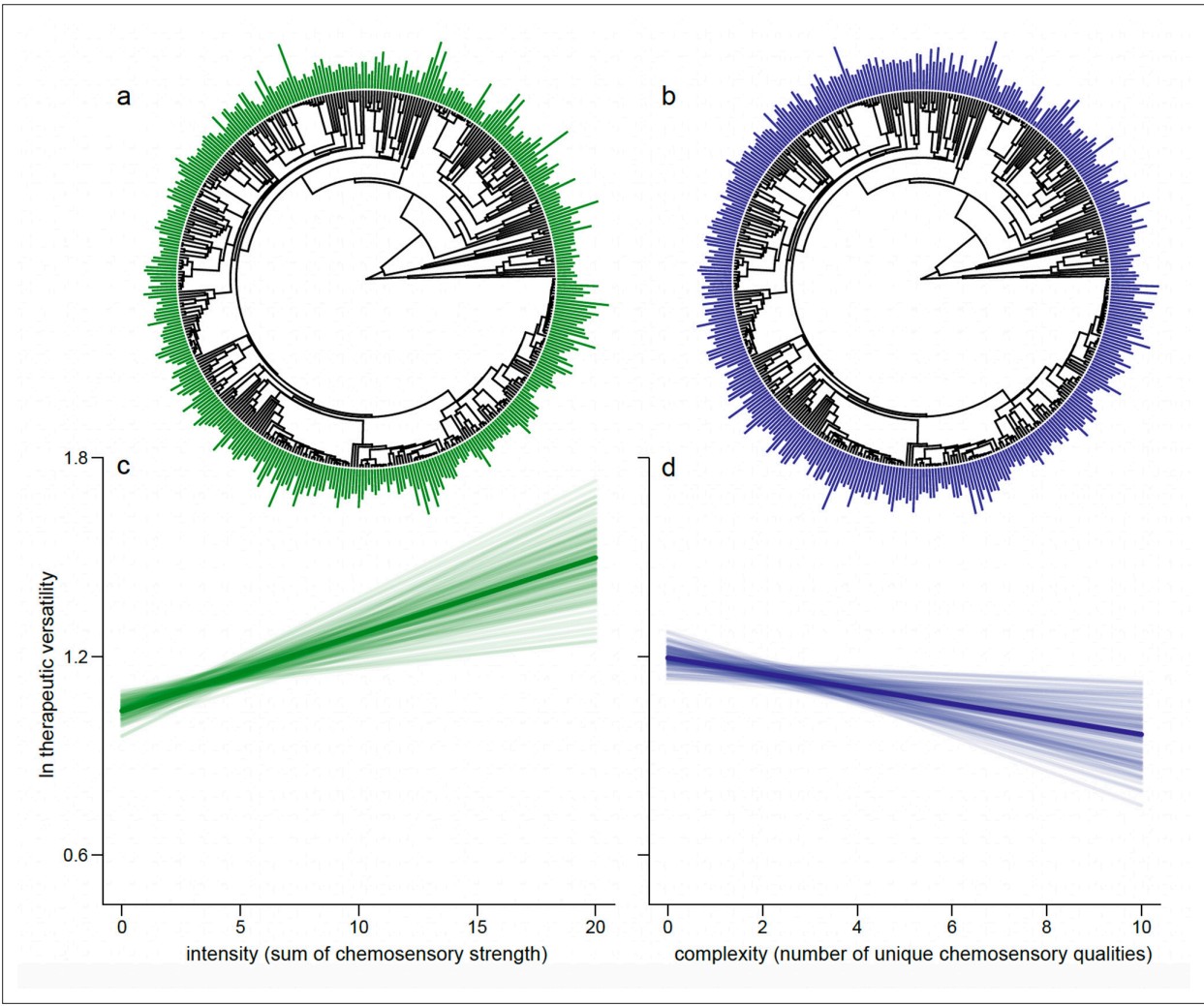

**Figure 2.** Drugs eliciting strong overall chemosensory perception (intense drugs) associated with relatively few perceived unique tastes (less complex) predict the number of categories of therapeutic use (therapeutic versatility). We show the distribution of intensity (**a**) and complexity (**b**) at the tips of the phylogenetic tree. As some species are represented by multiple parts, an average is shown here for representative purposes; each plant part is considered separately in the analysis. We plot the mean regression line as estimated from the parameter estimates of our Bayesian phylogenetic regression analyses (dark line) along with 100 random samples of the posterior distribution (faded lines). For graphical representation, the slopes are estimated holding other effects at their mean. We do this for both the significant positive association between intensity and therapeutic versatility (**c**, $p_x$ = <0.001) and the significant negative association between chemosensory complexity and therapeutic complexity (**d**, $p_x$ = 0.008).

The online version of this article includes the following figure supplement(s) for figure 2:

**Figure supplement 1.** The relationship between individual therapeutic uses and chemosensory intensity/complexity.

often perceived as weak. Panellists perceived between zero (e.g. *Anemone coronaria*, root) and ten (*Cinnamomum verum*, bark) individual qualities per drug. The mean number of therapeutic uses per botanical drug was 6.3 and the most frequently mentioned therapeutic use being 'gynaecology – abortion and menses' (272 use records) and the least frequently mentioned 'cardiac problems' (7 use records).

## Simple and intense chemosensory perception is associated with therapeutic versatility

The relationship between intensity of chemosensation (the sum of all chemosensory scores across all qualities, *Figure 2a*; see Methods) and therapeutic versatility is positive ($p_x < 0.001$; *Figure 2c*). On the other hand, chemosensory complexity (the number of taste and flavour qualities a botanical drug is scored for, *Figure 2b*; see Methods) is negatively associated with how many different out of the 25 categories of therapeutic use botanical drugs are used for (e.g. cardiac, gynaecological, humoral, henceforth referred to as 'therapeutic versatility'; $p_x = 0.026$; *Figure 2d*). Importantly, the multiple regression framework we use allows for each trait to covary whilst simultaneously predicting their underlying relationship with therapeutic versatility.

## Strength of specific chemosensory qualities predicts therapeutic versatility

All models we analyse have strong phylogenetic signal (modal $h^2 > 0.9$, see Methods) which may have led to misleading impressions about medicinal tastes and flavours in the past in analyses which did not account for such statistical non-independence. Indeed, while we do find a significant positive association between the strength of bitterness and therapeutic versatility ($p_x = 0.013$), the most versatile drugs were not only the bitter-tasting ones; contrary to popular concepts of bad-tasting medicine (*Mennella et al., 2013*). We also find that the perceived strengths of starchy ($p_x = 0.017$), musky ($p_x = 0.013$), sweet ($p_x = <0.018$), and soapy ($p_x = 0.002$) qualities were positively associated with therapeutic versatility while the intensity of perceived sourness was negatively associated with versatility ($p_x = <0.001$).

## Associations between specific chemosensory qualities and specific uses

All therapeutic uses (with the exception of 'gynaecology – other') are significantly associated with at least one specific chemosensory quality, whether positively (99 instances) or negatively (50 instances) and often with high magnitude of effect (*Figure 3*; *Supplementary file 5*). Thus, the perceived presence or absence of several specific qualities significantly confer the chemosensory perception of medicine (less than 5% of their estimated parameter distribution crosses zero, $p_x < 0.05$) with high magnitude of effect (*Figure 3*).

## Discussion

Unexpectedly, botanical drugs eliciting fewer but intense chemosensations were more versatile (*Figure 2*). People often associate complexity with intensity, and taste complexity is popularly interpreted with a higher complexity of ingredients (*Spence and Wang, 2018*). However, chemosensory perception is relative to exposed stimuli and simple tastes can be associated with complex chemistry when intense tastes mask weaker tastes, or when tastants are blended (*Breslin and Beauchamp, 1997*; *Green et al., 2010*). For example, starchy flavours or sweet tastes can be sensed when bitter and astringent antifeedant compounds are present below a certain threshold while salts enhance overall flavour by suppressing the perception of bitter tastants (*Breslin and Beauchamp, 1997*; *Johns, 1990*). On the other hand, combinations of different tastants or olfactory stimuli do not necessarily result in increased perceived complexity (*Spence and Wang, 2018*; *Weiss et al., 2012*). The relationship between intense chemosensation and versatility (*Figure 2*) is consistent with a perceived causal link between strong sensations and strong effects that is probably innate (*Ganchrow et al., 1983*; *Glendinning, 1994*). We also detected nuances in significance, and complete absence of significance across the relationships between individual therapeutic uses and complexity/intensity magnitudes for which we lack, however, more specific explanations (*Figure 2—figure supplement 1*).

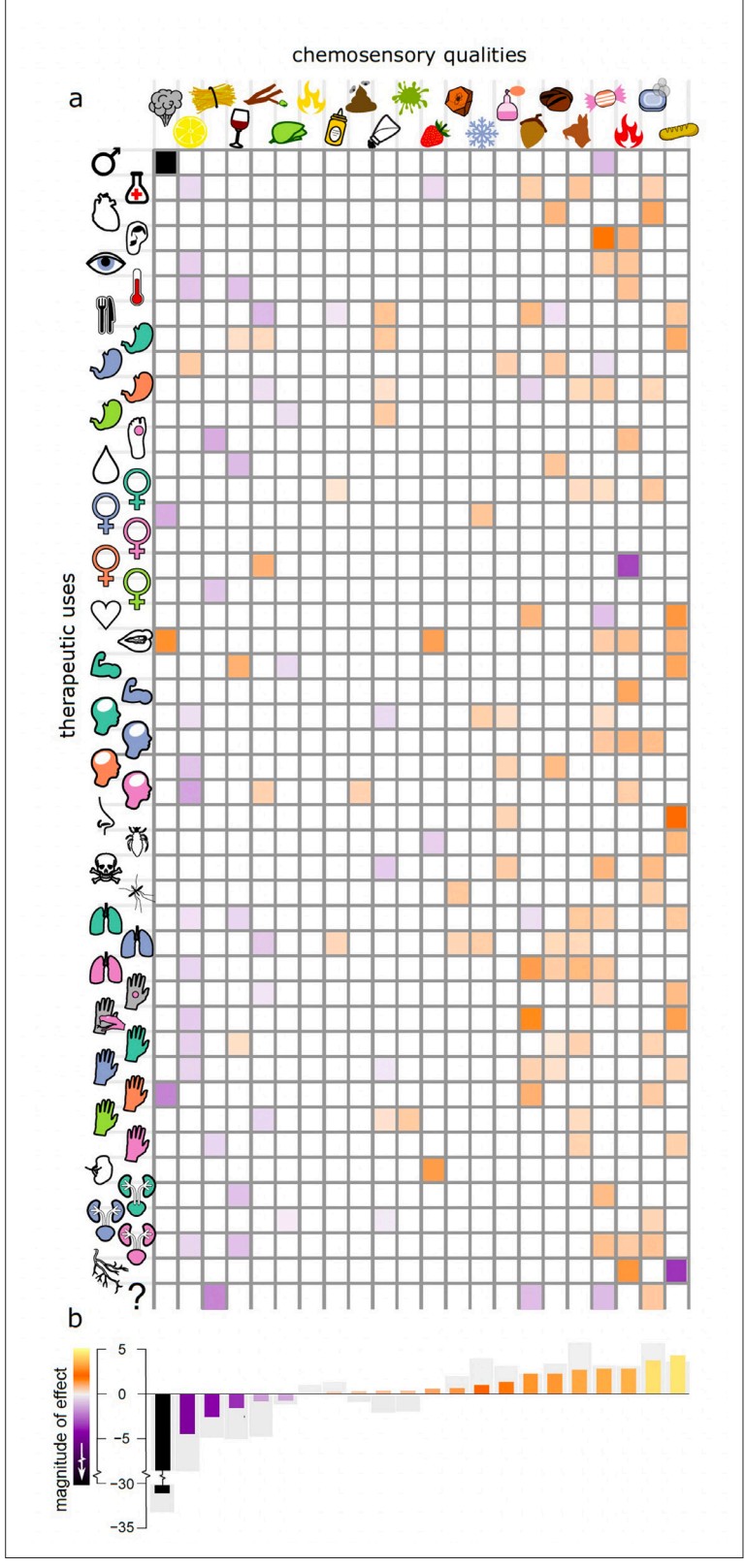

**Figure 3.** The magnitude of effect for 22 different specific chemosensory qualities across 46 different therapeutic uses. (a) A heat map showing the strength of the association between specific qualities (columns) and therapeutic uses (rows). Where we find a significant association between the strength of chemosensory quality and whether a drug is used for a particular therapeutic purpose in our phylogenetic binary (probit) response model, the

*Figure 3 continued on next page*

*Figure 3 continued*

corresponding cell is coloured according to its estimated parameter value; see colour scale in panel (**b**). All other variables are zeroed and coloured white for visualisation. (**b**) An overall 'magnitude of effect' is calculated for each chemosensory quality by summing the significant betas ($p_x < 0.05$, i.e. those displayed in the heat map above) across all different uses and plotted as bars coloured by their magnitude; see colour scale alongside axis. The sum of all betas regardless of significance are shown as grey bars. Note that the large parameter estimate for 'smoky' is owing to the fact that no botanical drug with any smoky quality (scored by any participant) is ever used for androgynous therapeutic purposes and thus is poorly estimated in that model (***Supplementary files 1, 2, 5, and 6***).

---

Low therapeutic versatility of sour tasting drugs (12 negative associations) may best be understood by considering the uses that this taste was negatively associated with in our PGLMMs (***Figure 3***). For example, the significant association between sour and treatment of coughs ($p_x = 0.038$) and other respiratory problems ($p_x = 0.042$; ***Figure 3***; ***Supplementary files 5 and 6***) might be related to coughing triggered by acid signalling in the airways (***Kollarik et al., 2007***). Moreover, acidic solutions and tissue acidosis are associated with inflammation and nociception while acid reflux is a common cause of chronic cough probably triggered by an oesophageal-bronchial reflex (***Irwin, 2006***). The negative associations of sour but also salty drugs with pain management ($p_x = 0.021$ and $0.022$, respectively; ***Figure 3***; ***Supplementary files 5 and 6***) are probably related to the avoidance of nociception known by personal experience to result from the contact between acidic or salty solutions and skin or mucous membrane lesions.

The therapeutic versatility of drugs perceived as sweet (13 positive associations) and starchy (12 positive associations) (***Figure 3***; ***Supplementary files 5 and 6***) is influenced by the many staples and edible fruits that are also used for medicine (e.g. protein-rich Fabaceae seeds were often perceived as having starchy tastes). The daily use of food items, generally with a low toxicity, has allowed for a more detailed comprehension of their postprandial effects, which arguably has led to a diversification of uses. The 96 botanical drugs specifically mentioned also for food (though there are more than 150 edible drugs in our dataset; ***Supplementary file 1***) show positive associations with starchy ($p_x = 0.005$), nutty ($p_x = 0.002$), and salty ($p_x = 0.001$) and negative associations with bitter ($p_x = 0.007$), woody ($p_x = 0.001$), and stinging ($p_x = 0.033$) chemosensory qualities.

Sweet sensations are known to induce analgesia (***Kakeda et al., 2010***) reflected here by a positive association of sweet-tasting drugs with the treatment of pain ($p_x = 0.032$). Both starchy ($p_x = 0.001$) and salty ($p_x = 0.002$) are associated with the treatment of diarrhoea and dysentery. Today, mixtures of glucose, starch, and electrolytes are used to treat secretolytic and inflammatory diarrhoea caused by bacterial enterotoxins (***Thiagarajah et al., 2015***; ***Binder et al., 2014***). These oral rehydration solutions exploit sodium/glucose co-transport by sodium/glucose co-transporter 1 in the small intestine and sodium absorption through sodium-hydrogen antiporter 2 stimulated by short chain fatty acids synthesised from starch by bacteria in the colon (***Thiagarajah et al., 2015***; ***Binder et al., 2014***). Drugs perceived as starchy are also positively associated with the treatment of topical ulcers ($p_x = 0.001$), skin infections ($p_x = 0.025$), other skin conditions ($p_x = 0.035$) and the topical treatment of venomous and non-venomous animal bites ($p_x = 0.001$) where plasters with lenitive properties might bring benefit. Moreover, fresh Fabaceae seeds contain protease inhibitors (***Birk, 1996***; ***Wink and Waterman, 2018***) such as trypsin and chymotrypsin inhibitors, which have the potential to inhibit bacterial proteases and control bacterial virulence during cutaneous infections (***Frees et al., 2013***; ***Culp and Wright, 2017***).

Bitter was the most frequently reported chemosensory quality and showed nine significant associations with therapeutic use – including the negative association with foods (***Figure 3***). Though many bitter compounds are toxic, not all bitter plant metabolites are (***Glendinning, 1994***; ***Drewnowski and Gomez-Carneros, 2000***; e.g. iridoids, flavonoids, glucosinolates, bitter sugars). In part, this may be the outcome of an arms race between plant defence and herbivorous mammals' bitter taste receptor sensitivities, resulting in the synthesis of metabolites capable of repelling herbivores and confounding the perception of potential nutrients by mimicking tastes of toxins. Here, poisons showed no association with bitter (perhaps because they were chosen so that they were unobtrusive and inconspicuous) but positive associations with aromatic ($p_x = 0.041$), sweet ($p_x = 0.022$), and soapy ($p_x = 0.025$) as well as a negative association with salty ($p_x = 0.046$) qualities (but see the discussion about humoral management).

Anti-inflammatory pharmacology of several bitter tasting compounds explains the positive associations with treatments of sciatica (p_x = <0.001; *Figure 3*). Bitter-tasting drugs to treat sciatica derive from several disparate lineages containing a range of different anti-inflammatory compound classes such as iridoids (*Viljoen et al., 2012*; *Ajuga*, *Centaurium*), sesquiterpene lactones (*Coricello et al., 2020*; *Artemisia*, *Inula*), triterpene saponines (*Gepdiremen et al., 2006*; *Wang et al., 2014*; *Citrullus*, *Ecballium*, *Leontice*), and cumarins (*Kirsch et al., 2016*; *Opopanax*, *Ruta*) (*Supplementary files 1 and 2*).

Generally, overarching explanations for versatility based on chemosensory perception are not feasible. However, we hypothesise that many of the associations we found (*Figure 3*; *Supplementary files 5 and 6*) are grounded in observed physiological effects mediated by chemesthesis and taste receptor pharmacology (*Chandrashekar et al., 2006*; *Palmer and Servant, 2022*). Cumulative evidence suggests that together with transient receptor potential channels (TRPs), extraoral taste receptors and ectopic olfactory receptors form a chemosensory network involved in homeostasis, disease response, and off-target drug effects (*Foster et al., 2014*; *An and Liggett, 2018*; *Kang and Koo, 2012*; *Hauser et al., 2017*; *Clark et al., 2012*). Bitter taste receptors, for instance, are located also in human airways where their activation mediates bronchodilation and the movement of motile cilia which leads to improved respiration and evacuation of mucus (*Shah et al., 2009*; *Deshpande et al., 2010*). Indeed, bitter showed positive associations with the treatment of breathing difficulties (p_x = 0.027) and other respiratory problems (p_x = 0.004). We find that drugs used to treat pain, including pain associated with breastfeeding (breast inflammation and lactation), are significantly more likely to have fresh and cooling qualities (p_x = 0.01 and 0.037, respectively). This is likely owing to interactions with TRP channels (*McKemy et al., 2002*; *Behrendt et al., 2004*; *Proudfoot et al., 2006*; *Memon et al., 2019*). Specifically, the TRPM8 channel mediates cold stimuli and is known to be activated by cooling essential oil constituents producing analgesic effects (*McKemy et al., 2002*; *Behrendt et al., 2004*; *Proudfoot et al., 2006*). Cooling menthol and eucalyptol have been shown to act as counterirritants inhibiting respiratory irritation caused by smoke constituents (*Willis et al., 2011*) explaining the positive association between drugs used to treat breathing difficulties with fresh and cooling flavours (p_x = 0.04). The strong positive association of burning and hot drugs with oral applications for treating musculoskeletal problems (p_x = 0.004) might be related to analgesic properties mediated by desensitising effects upon TRP channel interaction (*Marwaha et al., 2016*; *Aroke et al., 2020*). Burning and hot drugs are also positively associated with vascular problems (p_x = 0.042) including varices and haemorrhoids and can be explained with vasorelaxant effects transmitted by TRP channels on the endothelial cells (*Montell, 2005*; *Pires and Earley, 2017*). TRPV channels are also involved in kidney and urinary bladder functioning (*Montell, 2005*) and TRPV1-knockout mice urinate more frequently without voiding their bladder (*Birder et al., 2002*). Interactions with TRPV1 and other TRPV receptors might therefore explain the use of hot botanical drugs to treat other urinary problems including urinary retention, incontinence, kidney, and bladder problems (p_x = 0.047).

The concepts of humoral medicine may have derived from the observation that the appearance and consistency of bodily fluids during disease change and that they can be modified. The positive association of bitter taste with humoral management (p_x = 0.004) as well as with liver problems and jaundice (p_x = 0.002) is explained by uses intended to purge and resolve bitter bile and seems to include what we call choleretics and cholekinetics today, such as the roots of the great yellow gentian (*Gentiana lutea*) or the herb of the common vervain (*Verbena officinalis*). However, several of the plant drugs used for humoral management have strong emetic and purgative effects which are manifestations of their toxicity. Such remedies were abandoned along with humoral therapy and include drugs derived from squirting cucumber (*Ecballium elaterium*), castor bean (*Ricinus communis*), spurge (*Euphorbia* spp.), hellebore (*Helleborus* sp.), and white hellebore (*Veratrum* sp.). Astringent and woody qualities showed positive associations with the treatment of diarrhoea and dysentery (p_x = 0.029 and p_x = 0.019) where there is a need for controlling the flow of fluids and for drying them up. Similarly woody, often correlating with astringency (*Scalbert et al., 1989*), was also positively associated with the staunching of vaginal discharge (p_x = <0.001). Curiously, astringency showed positive associations with external applications for musculoskeletal conditions (p_x = 0.001) where drying internal bleeds and solidifying tissues and bones are needed. Negative associations of astringency emerged with conditions where the flux of humours, perspiration, and the evacuation of liquids and mucus are treatment strategies, such as in fever (p_x = 0.045), humoral management (p_x = 0.02), urinary

calculus ($p_x$ = 0.02), other urinary problems (retention, incontinence, kidney, and bladder problems, $p_x$ = <0.001) and cough ($p_x$ = 0.002). More obviously, the intake of salt aggravates fluid retention and oedema, and this is likely the reason why remedies for dropsy and oedema (diuretics) are negatively associated with salty tastes ($p_x$ = 0.023).

While chemosensory qualities of botanical drugs and their physiological effects help to explain the persistence of humoral medicine until the 19th century, our study highlights the potential of chemosensory pharmacology in the development of herbal medicines. Botanical drugs with observed chemosensory profiles not aligned with those of their therapeutic use might represent drugs working beyond chemosensory pharmacology. Conversely, associations of chemosensory qualities with therapeutic uses of botanical drugs may certainly overlap with the presence of metabolites with specific therapy-relevant constituents acting beyond chemosensory pharmacology. This is, for instance, the case with opium (dried milk sap of *Papaver somniferum*) indicated for 'cough' and perceived as bitter (*Figure 3*; *Supplementary files 1 and 2*) but at the same time containing alkaloids with central cough suppressant properties (*Dewick, 2001*). Besides the associations discussed, other therapeutically relevant chemosensory receptor functions might be involved in the associations currently lacking explanations. While we are aware that the limitation of this study is the impossibility to address all associations found and the dependence on published pharmacological data for the explanatory part, our approach demonstrates the power of phylogenetic methods applied to human behaviour and assessment of plant properties. We show that chemosensory perception connects ancient therapeutic knowledge with modern science, even though aetiologies were fundamentally different in Graeco-Roman times. Our study offers a blueprint for macroscale re-evaluation of pre-scientific knowledge and points towards a central role for taste and flavour in medicine.

## Methods

### Botanical fieldwork

A total of 700 botanical drugs (ca. 70% of the botanical drugs described in *DMM*) associated with 407 species were collected from the wild, cultivated in home gardens, or purchased from commercial sources between 2014 and 2016. Plant collection included at least one voucher per species and one or more bulk samples of the therapeutic plant part (botanical drug). Latin binomials follow https://powo. science.kew.org/. Botanical vouchers were deposited at the herbarium of the University of Geneva (G) and the herbarium of the National and Kapodistrian University of Athens (ATHU). Bulk samples for sensory analysis were dried at 40–60°C and stored in plastic containers. Collection permits were obtained from the Greek Ministry of Environment and Energy (6Ω8Κ4653Π8-ΑΚ7).

### Data extraction from *DMM*

Information about the therapeutic uses (use records) of botanical drugs were extracted from Matthioli's 1568 edition of Dioscorides' *De Materia Medica* (*Matthioli, 1970*; *Staub et al., 2016*). Botanical drugs were arranged into 46 therapeutic uses defined by organ, therapeutic function, disease, and symptom across 25 broader categories of use (*Figure 1*; *Supplementary files 1 and 4*). Each of the 700 botanical drugs was then assigned a binary variable defining whether it was recommended (1) or not (0) for each of the 46 therapeutic uses based on their descriptions in *DMM*. When the description in *DMM* permitted identification (Taxon ID; *Supplementary file 1*) only to genus level (due to ambiguities or the possibility that different closely related taxa were subsumed), we chose one species as the representative for the whole genus (*Supplementary file 1*: 'Taxon panel'). Unidentified taxa or taxa identified only to the family level were not considered in this analysis. The category 'food' includes only those drugs where edibility was specifically mentioned in *DMM* and by far not all those that are used as food. As a measure of therapeutic versatility, we summed up the total number of broader categories of therapeutic use each botanical drug was recommended for.

### Tasting panel

The chemosensory qualities of each of the 700 botanical drugs were assessed experimentally by trained human subjects using the sensory analysis technique Conventional Profiling (*International Standard, 1994*) performed in accordance with the Declaration of Helsinki (*World Medical Association, 2013*) and the ethics guidelines of the Institute of Food Science & Technology (*Institute of*

*Food Science and Technology, 2020*). The design of the tasting panel was approved by the Ethics Committee of the University Hospital of Cagliari (NP/2016/4522). Informed consent was obtained from all panellists. We used 22 chemosensory descriptors to represent taste and flavour perception (*Figure 1*; *Supplementary files 2 and 3*). The evaluation took place at the Hospital of *San Giovanni di Dio* (Cagliari, Italy) during a period of 5 months in 2016. The panel consisted of 11 healthy Caucasian volunteers (six males and five females; age 30.7 ± 7.4). Working language was Italian. Based on a random subset of 40 (out of 700) samples, four initial training sessions were held to establish a consensus regarding chemosensory terminology. Hedonic (e.g. 'good' or 'bad'), self-referential (e.g. 'minty' for mint), and infrequent descriptors were excluded to reduce subjectivity and to increase discriminatory power of the analysis. Synonymous and closely related descriptors (e.g. 'astringent' and 'tannic') were pooled to reduce attribute redundancy.

Samples (0.1–2 g of dried drug pieces) were labelled randomly (using a random number generator) with three-digit codes, assigned randomly and double-blinded (in case of common drugs impossible) to individual panellists. Not all drugs were tasted by all panellists while random distribution permitted that individual panellists were challenged with repeated samples. Samples were presented in identical 125 ml plastic containers at room temperature. The relatively big range in quantity of dispensed samples owes to toxicological concerns and practical reasons but in any case, were tailored to permit the perception of their chemosensory qualities. Panellists were instructed to chew the amount of sample necessary for chemosensory perception, to annotate perceived qualities and intensities and to spit out residues of samples, and finally rinse their mouth with drinking water. The breaks between tasting different samples depended on the persistence of chemosensory perception. For each sensory trial a separate evaluation sheet was used. Quality intensities were evaluated on a 4-point ordinal scale: absent (0), weak (1), medium (2), and strong (3). Overall, 3973 individual sensory trials were conducted, with an average of 361 ± 153 trials per panellist and 5.7 ± 1.3 trials per botanical drug. On average each panellist tasted 17.2 drugs per hour using 10.5 sessions (18 sessions in total) lasting approximately 2 hr each. From the ordinal data, two additional metrics were calculated to represent chemosensory *complexity* (the total number of different chemosensory qualities a botanical drug was assigned to as non-zero by each panellist) and chemosensory *intensity* (the sum of all chemosensory scores assigned across all qualities by each panellist).

## Phylogenetic tree

We used the species-level supertree of land plants to represent the relationships among the taxa extracted from *DMM* (*Zanne et al., 2014*). Some species were not found in the tree; in these cases, species were attached to the most recent common ancestor of the respective genus or, if absent, the family. Branch lengths of added taxa were set to retain ultrametricity. Tree manipulations were made in R (*R Development Core Team, 2016*) using the packages APE and PHYTOOLS (*Paradis et al., 2004*; *Revell, 2012*). The trees are represented in *Figure 2* alongside the chemosensory intensity and complexity data.

## Statistical procedure

In order to determine whether chemosensation and therapeutic uses were linked, we used two sets of phylogenetic linear mixed models implemented in a Bayesian framework within the R package MCMCglmm (*Hadfield, 2010*). The first set of models considered therapeutic versatility (number of categories of therapeutic use) modelled as a zero-truncated Poisson distributed response variable. We ran two models: (1) considering all 22 individual chemosensory qualities as ordinal-scale independent variables and (2) considering chemosensory intensity and complexity as independent variables. Together, these models tell us whether any particular chemosensory quality as well as whether intense or complex chemosensation predict how often a drug is used for any therapeutic purpose. We then ran a second set of probit models which considered each individual therapeutic use as a binary response. That is, we ran 46 models examining the relationship between individual chemosensory qualities and therapeutic uses (one for each use, where all qualities are included as independent variables) and an additional 46 to test the relationship between each therapeutic use and chemosensory intensity and complexity.

The significance of all parameters is assessed using the proportion of the posterior distribution of parameter estimates that cross zero ($p_x$). If a parameter has a substantial effect on a model, we expect

the distribution of parameter estimates to be shifted away from zero (i.e. $p_x < 0.5$). All models were run for a total of 500,000 iterations after burn-in, sampling every 5000.

Phylogenetic signal was calculated using heritability ($h^2$) which has an identical interpretation to Pagel's lambda. All models showed high heritability values ($h^2 > 0.9$) demonstrating the importance of species' shared ancestry ($h^2$ ranges between 0 and 1, and values close to 1 indicate very strong phylogenetic signal). In all models we included plant part, panellist ID, and the phylogenetic matrix as random effects. The inclusion of plant part and the phylogenetic matrix allowed for the fact that in some cases, multiple drugs were extracted from different parts of the same plant (e.g. *Vitis vinifera*; fruit, leaf, herb, seeds) whilst still accounting for the overall underlying evolutionary relationships. The inclusion of panellist ID removed the effect of any biased perception by single panellists across drugs as well as any differences among panellists for an individual drug. We also included the number of days between sample collection and when each sample was tasted, in order, to account for any degradation of samples over time that may have affected chemosensory perception. For all fixed factors in all models, we used largely uninformative priors (implementing a normal distribution with a mean of 0 and a variance of 1e10). Residual variance was estimated using an inverse gamma prior (V=1, nu = 0.002). We used parameter-expanded priors (roughly uniform standard deviations) for all random effects including phylogenetic variance (*Hadfield and Nakagawa, 2010*).

## Acknowledgements

We are thankful for support and help to Micaela Morelli, Marco Pistis, Simona Scalas, Luigi Raffo, Paolo Mura, Catina Chilotti, Chris Venditti, the panellists, Ettore Casu, Valentina Cazzaniga, Stefania Fortuna, Alessandro Riva, and the Ethics Committee of the University Hospital of Cagliari.

## Additional information

### Funding

| Funder | Grant reference number | Author |
| --- | --- | --- |
| Seventh Framework Programme | 606895 | Marco Leonti |

The funders had no role in study design, data collection and interpretation, or the decision to submit the work for publication.

### Author contributions

Marco Leonti, Conceptualization, Resources, Data curation, Formal analysis, Supervision, Funding acquisition, Investigation, Methodology, Writing – original draft, Project administration, Writing – review and editing; Joanna Baker, Software, Formal analysis, Validation, Investigation, Visualization, Methodology, Writing – original draft, Writing – review and editing; Peter Staub, Conceptualization, Data curation, Investigation, Methodology; Laura Casu, Resources, Data curation, Methodology, Project administration; Julie Hawkins, Supervision, Investigation, Writing – original draft, Writing – review and editing

### Author ORCIDs

Marco Leonti ⓘ https://orcid.org/0000-0002-4726-9758
Joanna Baker ⓘ http://orcid.org/0000-0003-4904-6934
Peter Staub ⓘ https://orcid.org/0000-0003-4875-7064
Laura Casu ⓘ https://orcid.org/0000-0001-6480-8680
Julie Hawkins ⓘ http://orcid.org/0000-0002-9048-8016

### Ethics

The design of the tasting panel was approved by the Ethics Committee of the University Hospital of Cagliari (NP/2016/4522). Informed consent was obtained from all panellists.

Reviewer #1 (Public Review): https://doi.org/10.7554/eLife.90070.3.sa1

Reviewer #2 (Public Review): https://doi.org/10.7554/eLife.90070.3.sa2
Author Response https://doi.org/10.7554/eLife.90070.3.sa3

## Additional files

### Supplementary files

• Supplementary file 1. Sampled botanical drugs and their uses *ex* Matthioli (1568).

• Supplementary file 2. Sampled botanical drugs and their chemosensory qualities (panel data).

• Supplementary file 3. Codes and the short descriptions of the 22 chemosensory qualities to represent tastes and flavours.

• Supplementary file 4. Codes and the short descriptions of the therapeutic uses.

• Supplementary file 5. Heat map showing the strength of the association between chemosensory quality and therapeutic use.

• Supplementary file 6. Significance of individual parameter estimates from the models of individual chemosensory quality against individual therapeutic use calculated as the proportion of the parameter estimate crossing zero, $p_x$.

• MDAR checklist

### Data availability

All data generated or analysed during this study are included in the manuscript and supporting files (*Supplementary files 1–6*).

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
