## [Editor Report · eLife assessment]

This **valuable** study links the "taste" of botanicals to their application as medicines used by the ancient Greco-Roman society. The authors used phylogenetic linear mixed models in a Bayesian framework to test the relationships between taste qualities, intensities, complexities, and therapeutic use. The evidence supporting the conclusions is **solid**, although there is a minor weakness concerning the somewhat inconsistent method of botanical preparation and presentation to the taster panelists; subjective bias and robustness of the participants' responses might have been overlooked. The study may be of broad interest to pharmacologists and scientists working on drug discovery, particularly those interested in natural products.

---

## [Referee Report · Reviewer #1 (Public Review)]

Summary: The authors explored correlations between taste features of botanical drugs used in ancient times and therapeutic uses, finding some potentially interesting associations between intensity and complexity of flavors and therapeutic potential, plus some more specific associations described in the discussion section. I believe the results could be of potential benefit for the drug discovery community, especially for those scientists working in the field of natural products.

Strengths:

Owing to its eclectic and somehow heterodox nature, I believe the article might be of interest for a general audience. In fact, I have enjoyed reading it and my curiosity was raised by the extensive discussion.

The idea of revisiting a classical vademecum with new scientific perspectives is quite stimulating.

The authors have undertaken a significant amount of work, collecting 700 botanical drugs and exploring their taste and association with known uses via eleven trained panellists.

Weaknesses:

I have some methodological concerns. Robustness in the panelists' perceptions has not been addressed, and not every panellist tasted every drug because of time constrains. The breaks between tasting different samples was not standardized, and depended on the persistence of chemosensory perception, possibly also due to time constraints.

---

## [Referee Report · Reviewer #2 (Public Review)]

Summary:

This is an unusual, but interesting approach to link the "taste" of plants and plant extracts to their therapeutic use in ancient Graeco-Roman culture. The authors used a panel of 11 trained tasters to test ~700 different medicinal plants and describe them in terms of 22 "taste" descriptors. They correlated these descriptors with the plant's medical use as reported in the De Materia Medica (DMM 1st Century, CE). Correcting for some of the plants' evolutionary phylogenetic relationships, the authors found that taste descriptors along with intensity measures were correlated with the "versatility" and/or a specific therapeutic use of the medicine. For example, simple but intense tastes were correlated with versatility of a medicine. Specific intense tastes were linked to versatility while others were not; intense bitter, starchy, musky, sweet, cooling and soapy were associated with versatility, but sour and woody were negatively associated. Also some specific tastes could be associated with specific uses - both positive and negative associations. Some of these findings make sense immediately, but others are somewhat surprising, and the authors propose some links between taste and medicinal use (both historical and modern use) in the discussion. The authors state that this study allows for a re-evaluation of pre-scientific knowledge, pointing toward a central role for taste in medicine.

Strengths:

The real strength of this study is the novelty of this approach - using modern day tasters to evaluate ancient medicinal plants to understand the potential relationships between taste and therapeutic use, lending some support to the idea that the "taste" of a medicine is linked to its effectiveness as a treatment.

Weaknesses:

Because of the limitations of time and the type of botanicals being tested, there is an inherent difficulty in assessing taste intensity. However, because these botanicals are tested by multiple panelists and sometimes tested repeatedly by individual panelists, this helps support the author's analyses.

---

## [Author Response]

The following is the authors’ response to the original reviews.

**Public Reviews:**

**Reviewer #1 (Public Review):**
Summary:The authors explored correlations between taste features of botanical drugs used in ancient times and therapeutic uses, finding some potentially interesting associations between intensity and complexity of flavors and therapeutic potential, plus some more specific associations described in the discussion sections. I believe the results could be of potential benefit to the drug discovery community, especially for those scientists working in the field of natural products.Strengths:Owing to its eclectic and somehow heterodox nature, I believe the article might be of interest to a general audience. In fact, I have enjoyed reading it and my curiosity was raised by the extensive discussion.The idea of revisiting a classical vademecum with new scientific perspectives is quite stimulating.The authors have undertaken a significant amount of work, collecting 700 botanical drugs and exploring their taste and association with known uses via eleven trained panelists.Weaknesses:I have some methodological concerns. Was subjective bias within the panel of participants explored or minimized in any manner?

Yes, in all models we included ‘panellist’ as a random effect and therefore any biased perception by a single panellist across drugs or differences among panellists for an individual drug was accounted for. We now make this clearer in our methods.

Were the panelists exposed to the drugs blindly and on several occasions to assess the robustness of their perceptions?

The study was double blind, but blinding was not possible with the more well-known drugs (e.g., almonds, walnuts, thyme, mint). A random number generator was used to assign the drugs to the panellists, and according to the random distribution, some drugs were presented to the same panellist more than once. Robustness of panellists’ perception was not assessed specifically. We have added some text to the methods to clarify.

Judging from the total number of taste assessments recorded and from Supplementary Material, it seems that not every panelist tasted every drug. Why?

Because there were many drugs and panellists had time constraints. Overall, 3973 individual sensory trials were conducted, with an average of 361±153 trials per panellist and 5.7±1.3 trials per botanical drug.

It may be a good idea to explore the similarity in the assessments of the same botanical drug by different volunteers. If a given descriptor was reported by a single volunteer, was it used anyway for the statistical analysis or filtered out?

All responses were used as reported by the panellists, including potential ‘outliers’. As described above, the inclusion of ‘panellist’ as a random effect means that if one individual gave an unusual description of a particular drug in comparison to other individuals, this would be less impactful on any parameter estimates.

The idea of "versatility" is repeatedly used in the manuscript, but the authors do not clearly define what they call "versatile".

In line with suggestions made by reviewers, we have slightly adjusted the definition of therapeutic versatility and have now clearly defined the term on first use. Here, we define therapeutic versatility as the number of therapeutic ‘categories’ a drug is used for (the 25 broad categories are represented by shared iconography in Figure 1). Our revised results include analyses using this definition – which are qualitatively identical to our previous results which defined versatility using the 46 individual therapeutic uses.

The introduction should be expanded. There are plenty of studies and articles out there exploring the evolution of bitter taste receptors, and associating it with a hypothetical evolutionary advantage since bitter plants are more likely to be poisonous.

We agree. Bitter is arguably the most frequent chemosensory attribute of plants and botanical drugs perceived by humans. Our data shows that ‘poisons’ are not associated with bitterness but positively with ‘aromatic’, ‘sweet’ and ‘soapy’ – and negatively with ‘salty’ qualities.

We have added this paragraph to the introduction:

"The perception of taste and flavour (a combination of taste, smell and chemesthesis) here also referred to as chemosensation, has evolved to meet nutritional requirements and are particularly important in omnivores for seeking out nutrients and avoiding toxins (Rozin and Todd, 2016; Breslin, 2013; Glendinning, 2022). The rejection of bitter stimuli has generally been associated with the avoidance of toxins (Glendinning, 1994; Lindemann, 2001; Breslin, 2013) but to date no clear relationship between bitter compounds and toxicity at a nutritionally relevant dose could be established (Glendinning, 1994; Nissim et al., 2017). While bitter tasting metabolites occurring in fruits and vegetables have been linked with a lower risk for contracting cancer and cardiovascular diseases (Drewnoswski and Gomez-Carneros, 2000) the avoidance of pharmacologically active compounds is probably the reason why many medicines, including botanical drugs, taste bitter (Johns, 1990; Mennella et al., 2013)."

And expanded in the discussion:

"Though many bitter compounds are toxic, not all bitter plant metabolites are (Glendinning, 1994; Drewnoswski and Gomez-Carneros, 2000; e.g., iridoids, flavonoids, glucosinolates, bitter sugars). In part, this may be the outcome of an arms race between plant defence and herbivorous mammals’ bitter taste receptor sensitivities, resulting in the synthesis of metabolites capable of repelling herbivores and confounding the perception of potential nutrients by mimicking tastes of toxins. Here, poisons showed no association with bitter but positive associations with aromatic (px = 0.041), sweet (px = 0.022) and soapy (px = 0.025) as well as a negative association with salty (px = 0.046) qualities."

Since plant secondary metabolites are one of the most important sources of therapeutic drugs and one of their main functions is to protect plants from environmental dangers (e.g., animals), this evolutionary interplay should be at least briefly discussed in the introductory section.

This is now referred to in the introduction as well as in the discussion.

Since the authors visit some classical authors, Parecelsus' famous quote "All things are poison and nothing is without poison. Solely the dose determines that a thing is not a poison" may be relevant here. Also note that some authors have explored the relationship between taste receptors and pharmacological targets (e.g., Bioorg Med Chem Lett. 2012 Jun 15;22(12):4072-4).

We agree that pharmacologic action is determined by the dose. We now refer to the dose in the introduction: “…to date no clear relationship between bitter compounds and toxicity at a nutritionally relevant dose could be established (Glendinning, 1994; Nissim et al., 2017)”.

We are aware of the fact that several authors have explored the relationship between taste receptors as targets and their similarity with other targets. We use many examples from the literature to explain our data. Our analysis did, however, not highlight any association between sweet tastes and epilepsy (as reported in Bioorg Med Chem Lett. 2012 Jun 15;22(12):4072-4). We are not able to explain all associations, and we acknowledge that there may be more associations between chemosensory receptors and therapeutic effects than those found and discussed here.

**Reviewer #2 (Public Review):**
Summary:This is an unusual, but interesting approach to link the "taste" of plants and plant extracts to their therapeutic use in ancient Graeco-Roman culture. The authors used a panel of 11 trained tasters to test ~700 different medicinal plants and describe them in terms of 22 "taste" descriptors. They correlated these descriptors with the plant's medical use as reported in the De Materia Medica (DMM 1st Century, CE). Correcting for some of the plants' evolutionary phylogenetic relationships, the authors found that taste descriptors along with intensity measures were correlated with the "versatility" and/or specific therapeutic use of the medicine. For example, simple but intense tastes were correlated with the versatility of a medicine. Specific intense tastes were linked to versatility while others were not; intense bitter, starchy, musky, sweet, cooling, and soapy were associated with versatility, but sour and woody were negatively associated. Also, some specific tastes could be associated with specific uses - both positive and negative associations. Some of these findings make sense immediately, but others are somewhat surprising, and the authors propose some links between taste and medicinal use (both historical and modern use) in the discussion. The authors state that this study allows for a re-evaluation of pre-scientific knowledge, pointing toward a central role of taste in medicine.Strengths:The real strength of this study is the novelty of this approach - using modern-day tasters to evaluate ancient medicinal plants to understand the potential relationships between taste and therapeutic use, lending some support to the idea that the "taste" of a medicine is linked to its effectiveness as a treatment.Weaknesses:While I find this study very interesting and potentially insightful into the development and classification of certain botanical drugs for specific medicinal use, I would encourage the authors to revise the manuscript and the accompanying figures significantly to improve the reader's understanding of the methods, analyses, and findings. A more thorough discussion of the limitations of this particular study and this general type of approach would also be very important to include.

Figures were revised, one deleted (former Fig. 3), and another one put to the supplementary (former Fig. 4, now Figure supplement 1). We now acknowledge limitations in the final paragraph.

The metric of versatility seems somewhat arbitrary. It is not well explained why versatility is important and/or its relationship with taste complexity or intensity.

We have modified the definition of versatility in line with reviewers’ comments. We have provided a detailed explanation of this in our response to reviewer #1 but for ease of reference, we paste this again here:

Here, we define therapeutic versatility as the number of therapeutic ‘categories’ a drug is used for (the 25 broad categories are represented by shared iconography in Figure 1). Our revised results include analyses using this definition – which are qualitatively identical to our previous results which defined versatility using the 46 individual therapeutic uses.

The importance of versatility was not the focus but the impact of taste intensity and complexity on versatility. We hypothesize that associations between perceived complexity and intensity of chemosensory qualities with versatility of botanical drug use provides insights into the development of empirical pharmacological knowledge and therapeutic behaviour (now included in the introduction).

Similarly, the rationale for examining the relationships between individual therapeutic uses and taste intensity/complexity is not well explained, and given that a similar high intensity/low complexity relationship is common for most of the therapeutic uses, it restates the same concepts that were covered by the initial versatility comparison.

The examination of the relationships between individual therapeutic uses and taste intensity/complexity fine-tunes the overall analysis and shows that this concept is applicable in general. However, in general, the reviewer is correct, and this is not our main focus. We therefore shifted the analysis including the figure to the supplementary material and state in the discussion: “We also detected nuances in significance, and complete absence of significance across the relationships between individual therapeutic uses and complexity/intensity magnitudes for which we lack, however, more specific explanations (Figure supplement 1).

There are multiple issues with the figures - the use of icons is in many cases counterproductive and other representations are not clear or cause confusion (especially Figure 3).

We have excluded former Fig. 3. Otherwise, the use of iconography is to facilitate graphical representation and cross-referencing between figures without over-cluttering. We provide all text and numeric values in the supporting information if individual detail is required.

The phylogenetic information about the botanicals is missing. Also missing is any reference/discussion about how that analysis was able to disambiguate the confounding effects of shared uses and tastes of drugs from closely related species.

This is explained in the methods (sections: ‘Phylogenetic tree’ and ‘statistical procedure’). We highlight that all models showed high heritability which means that shared ancestry has a statistical influence on the model. The trees themselves are now represented in our modified Figure 2.

**Reviewer #1 (Recommendations For The Authors):**
Besides the points already covered in my public review, I believe it would be interesting to assess and discuss the differences between the category "food" (how many drugs were allocated there?) and the drugs used for therapeutic purposes. In this manner, the food category could serve as a retrospective negative control to test the authors' hypotheses. Does the food category include drugs of weak flavor? Does it include drugs of complex flavor?

All drugs in this database are associated with therapeutic uses. Only 96 are specifically mentioned to be also used as food while in total at least 152 are also used as food (many of the most obvious food drugs are not labelled as such in DMM). It is difficult to use the food category as a negative control (for testing whether food drugs have weaker tastes), because spices are included in the food category. If at all, only staples should be used for such an analysis. But this would be another study.

In the context of the present analyses, we do agree that there is interest and so we have therefore added a small section to our manuscript: The 96 botanical drugs specifically mentioned also for food (though there are more than 150 edible drugs in our dataset; Supplementary file 1) show positive associations with starchy (px = 0.005), nutty (px = 0.002) and salty (px = 0.001) and negative associations with bitter (px = 0.007), woody (px = 0.001) and stinging (px = 0.033) tastes and flavours.

Please replace "plant defence" with "plant defense".

Currently the whole MS is formatted BE. We are happy to revise on the basis of editorial policy.

**Reviewer #2 (Recommendations For The Authors):**
1. I would encourage replacing "taste" with "flavor" throughout the manuscript and in the title because this paper addresses "taste here defined as a combination of taste, odour and chemesthesis" which essentially is the definition of flavor, and should not be simplified to taste. Flavor is the more precise word, and there is no need to confuse readers by defining "taste" in this way when taste means just the gustatory aspect of flavor.

We now define flavour as a combination of taste, smell and chemesthesis and use ‘taste’ when referring to a specific taste quality. We use the term ‘chemosensory’ (perception, quality) and chemosensation for addressing the perception of both, taste and flavour qualities together. The abstract now reads: “The perception of taste and flavour (a combination of taste, smell and chemesthesis) here referred to as chemosensation, enables animals to find high-value foods and avoid toxins.”

We prefer to leave the title as it is in accordance with standard books (e.g., “Pharmacology of Taste” by Palmer and Servant) which address all kinds of chemosensory interactions and the fact that we’ve conducted a ‘tasting panel’ (and not a ‘flavour panel’), and because flavour as a concept is only used in English (and also there not consistently, with ‘taste’ being the preferred term used by English native speakers for describing perception where in a strict sense, ‘flavour’ would be the correct term, see Rozin P. "Taste-smell confusions" and the duality of the olfactory sense. Percept Psychophys. 1982 Apr;31(4):397-401) and maybe also in French.

1. Methods - A much more detailed description of how the samples were prepared for the taste tests is needed. Were they sampled as a dry powder?No, they were sampled as dried pieces. We have added more information to our methods section to clarify.

Why is there such a big range in the amount provided (.1 to 2 g)?Because certain drugs are highly toxic (aconitum, opium) we could only provide a relatively small amount (that still permitted the perception of taste qualities). For practical reasons, half a walnut was dispensed. We have added more information to our methods section to clarify.

Also "Panelists were instructed to spit, rinse their mouth with drinking water and to take a break before tasting the next sample" This seems more likely that the samples were dissolved in a liquid if they were spitting and rinsing, but this is not clear. Also - take a break for how long between samples?

Panellists were instructed to chew the amount of sample necessary for taste perception, to annotate their perception, and to spit out residues of samples and finally rinse their mouth with drinking water. The breaks between tasting different samples depended on chemosensory persistence. We have added more information to our methods section to clarify.

How many samples were tested per day?

The number of tasted samples was different from panelist to panelist and depending on available time frames. On average each panellist tasted 17,2 drugs per hour using 10.5 sessions (18 sessions in total) lasting approximately two hours each. We have added more information to our methods section to clarify.

Did individual panelists get repeated samples?

Random distribution permitted that individual panellists were challenged also with repeated samples. We have added more information to our methods section to clarify.

1. Methods - Phylogenetic tree - Where is the output of this tree? It should be included in the figures and referred to in the results/discussion where the authors claim that they have been able to disambiguate phylogenetic closeness with taste and medicinal use.

We did not ‘build’ a phylogenetic tree, rather we modified an existing one. Therefore, the wording of that section in the methods has been adjusted for clarity. We refer to the tree in the results pertaining to phylogenetic relatedness by explicitly quantifying the extent of phylogenetic signal using the widely used heritability (h2) statistic. This means that shared ancestry has a statistical influence on the model. We have also added to our Figure 2 representations of the phylogenetic tree we used in our analysis, limited to the species for which we have data, also displaying the data (in this case, intensity and complexity) at the tips.

1. Taste intensity ratings should be better explained. Since the panelists are evaluating different amounts of samples (.1 to 2g) wouldn't the intensity of taste also depend on the amount of the substance?

The panelists were not told to introduce all the sample into their mouth but just enough to perceive the taste qualities clearly (explanation given in methods). E.g.: one black pepper corn is normally enough to perceive the taste and flavour of pepper while the same amount of hazelnut would be insufficient.

Or is this measure a relative value - "woodiness" vs "sourness" for example within the sample is strong/weak?

Chemosensation and sensory perception in general is always relative. (For instance, currently I can hear the birds singing outside. Was there music playing in my room I wouldn’t be able to hear them).

Because of this - are samples with strong tastes less likely to seem complex because the intensity of one stimulus masks the other?

Yes, we argue that drugs with strong tastes/flavours are less likely be perceived as being complex (fewer individual qualities perceived), arguably because strong stimuli overshadow weaker ones. We currently address this in the discussion and have made some modifications in line with the below comment.

This issue was presented briefly in the discussion when addressing the finding that samples with intense, but fewer tastes were more versatile, but this was highly confusing.The authors presented both sides of the problem without referring to any of their own experiments to resolve the issue, or to highlight this as a potential limitation of the study at hand.

Yes, stronger tastes mask weaker tastes which addresses both sides of the problem.

We have modified the first paragraph of the discussion to make this clearer.

It now reads: "Unexpectedly, botanical drugs eliciting fewer but intense chemosensations were more versatile (Fig. 2). People often associate complexity with intensity, and taste complexity is popularly interpreted with a higher complexity of ingredients (Spence, and Wang, 2018). However, simple tastes can be associated with complex chemistry when intense tastes mask weaker tastes, or when tastants are blended (Breslin and Beauchamp, 1997; Green et al., 2010). For example, starchy flavours or sweet tastes can be sensed when bitter and astringent antifeedant compounds are present below a certain threshold while salts enhance overall flavour by suppressing the perception of bitter tastants (Breslin and Beauchamp, 1997; Johns, 1990). On the other hand, combinations of different tastants or olfactory stimuli do not necessarily result in increased perceived complexity (Spence and Wang, 2018; Weiss et al., 2012)."

It would be useful to understand the parameters a bit more - a data visualization of the relationships of intensity and complexity across all samples would be a welcome addition to Figure 2.

Shared ancestry has a statistical influence on the model. We have now also added to our Figure 2 representations of the phylogenetic tree we used in our analysis, limited to the species for which we have data, also displaying the data (in this case, intensity and complexity) at the tips.

1. "Therapeutic Versatility" is a measure of how many different therapeutic uses a given botanic is listed in the DMM. This is one of the primary comparisons of this study, but the authors do not provide much of a rationale for using this metric. Also, there are 46 therapeutic uses, but many are interrelated such as gastric, gynecology, muscle, neurological, respiratory, skin, and kidney. It is not clear in my reading of the methods if this was also treated in some type of "phylogeny" as well or not. I would assume a real therapeutic versatility metric should be higher for something used for cough, ulcers, gout, and menses rather than something that was used for 4 different, but skin-related complaints.

The reviewer is correct, and we appreciate this comment. We have modified the definition of versatility in line with the suggestions laid out here. We have provided a detailed explanation of this in our public responses but for ease of reference, we paste this again here:

Here, we define therapeutic versatility as the number of therapeutic ‘categories’ a drug is used for (the 25 broad categories are represented by shared iconography in Figure 1). Our revised results include analyses using this definition – which are qualitatively identical to our previous results which defined versatility using the 46 individual therapeutic uses.

We repeated our original ‘versatility’ analyses using the 25 broader categories rather than the 46 individual uses. The results remained largely the same.

1. Use of icons/pictorial representations in figures. Overall, the use of icons is not necessary - words could be used, and then readers would not need to keep going back and forth to the key in Figure 1 to identify the taste/use. I am very confused by Figure 3. How is the strength of taste shown in this figure? The use of the balance is a confusing representation since I don't associate strength/intensity with weight. Also there are specific tastes that are used more, and others that are used less (but the numbers of those are also more/less). I do not think this figure accomplishes the goal of relaying these findings.

Whilst we agree that iconography is not strictly necessary, we think it is a good way of graphically representing the results without over-crowding the figures or introducing text sizes too small to read in print. All values are provided in the supporting information if any individual detail is required.

We have decided on the basis of these comments to exclude former Fig. 3 and (Figure supplement 1). We hope that the removal of this figure and clearer signposting towards the text and numerical tables in the supplementary information alleviates the reviewer’s concerns.

1. Similarly, figure 4 is unclear. This could be better represented in a table with words and p values listed. But a larger issue is that this shows essentially the same overarching relationship across the therapeutic use cases - high intensity, low complexity. Only the pink kidney (other?) case differs from this pattern. In the discussion, several therapeutic uses are discussed that could need intense tasting medicine - but these are not related directly back to the relationships shown in Figure 4.

Yes, we agree with the reviewer and have now moved Fig. 4 to the supplementary (Figure supplement 1)